# Correlation Analysis of In-Vehicle Sensors Data and Driver Signals in Identifying Driving and Driver Behaviors

**DOI:** 10.3390/s23010263

**Published:** 2022-12-27

**Authors:** Lucas V. Bonfati, José J. A. Mendes Junior, Hugo Valadares Siqueira, Sergio L. Stevan

**Affiliations:** 1UTFPR, Graduate Program in Electrical (PPGEE), Federal Technological University of Parana, Ponta Grossa 84017-220, Brazil; 2UTFPR, Graduate Program in Electrical and Computer Engineering (CPGEI), Federal Technological University of Parana, Curitiba 80230-901, Brazil

**Keywords:** driver behavior analysis, feature extraction, CAN network, classification, pattern recognition

## Abstract

Today’s cars have dozens of sensors to monitor vehicle performance through different systems, most of which communicate via vehicular networks (CAN). Many of these sensors can be used for applications other than the original ones, such as improving the driver experience or creating new safety tools. An example is monitoring variables that describe the driver’s behavior. Interactions with the pedals, speed, and steering wheel, among other signals, carry driving characteristics. However, not always all variables related to these interactions are available in all vehicles; for example, the excursion of the brake pedal. Using an acquisition module, data from the in-vehicle sensors were obtained from the CAN bus, the brake pedal (externally instrumented), and the driver’s signals (instrumented with an inertial sensor and electromyography of their leg), to observe the driver and car information and evaluate the correlation hypothesis between these data, as well as the importance of the brake pedal signal not usually available in all car models. Different sets of sensors were evaluated to analyze the performance of three classifiers when analyzing the driver’s driving mode. It was found that there are superior results in classifying identity or behavior when driver signals are included. When the vehicle and driver attributes were used, hits above 0.93 were obtained in the identification of behavior and 0.96 in the identification of the driver; without driver signals, accuracy was more significant than 0.80 in identifying behavior. The results show a good correlation between vehicle data and data obtained from the driver, suggesting that further studies may be promising to improve the accuracy of rates based exclusively on vehicle characteristics, both for behavior identification and driver identification, thus allowing practical applications in embedded systems for local signaling and/or storing information about the driving mode, which is important for logistics companies.

## 1. Introduction

According to the National Highway Traffic Safety Administration (NHTSA), about 9% of fatal crashes/collisions in the United States in 2017 involved driver distraction [1]. Interactions with multimedia systems or food are examples of activities contributing to the driver taking their eyes off the road and the signals around them [1]. In addition to these distracting actions, the driver’s behavior pattern while driving can be one of the causes of crashes. It is an essential factor because about 78% of drivers have already shown at least some aggressive behavior while driving (in 1 year) [2].

Using devices for the safety and prevention of automotive collisions is common. Classic examples are seat belts [3], airbags, anti-lock braking systems (ABS) [4], and electronic stability control (ESC) [5]. These devices can be mixed with complex monitoring systems such as advanced driver assistance [6] for improving safety and preserving the integrity of drivers and passengers.

However, various sensors and devices can be implemented and developed to complement drivers’ and passengers’ safety for crash prevention. Sensors on the steering wheel that detect the presence of the hands [7], smartphones with algorithms to analyze the driver’s direction [8], inertial sensors to check the driver’s driving [9,10], cameras to check the driver’s facial expression [11], and foot movements to predict footsteps [12] are examples of assistive devices to increase safety. Behavior monitoring can involve combining biopotentials such as ECG [13], EEG, and electrodermal activity [14] to assess driver behavior, fatigue, and stress levels. There is a tendency to use sensor fusion to improve these tasks, combining data obtained by the controlled area network (CAN) bus and external sensors to the vehicle and driver. For example, one can note using the position sensor on the accelerator, a camera, and data from the CAN bus to assess how critical the situation is and the driver’s intention to intervene in direction [15]. Another example is combining the data from the CAN bus, microphones, and cameras to identify everyday driving sections and where activities are carried out along the way [12].

Regarding interaction with the vehicle, it is noteworthy that only the accelerator pedal is constantly monitored by the CAN bus, while other pedals (brake and clutch pedals) have not been treated in the literature as continuous information usually monitored in passenger vehicles. Moreover, when this information is used, the data obtained are not standardized, making their analysis difficult [16]. Although the information on vehicle speed and revolutions per minute (RPM), accelerator pedal position, wheel braking (ABS), and brake oil pressure is available on the CAN bus, in most cars, there is no monitoring of this position pedal in the function of the driver’s action [17]. Furthermore, it is still difficult to obtain specific data from the CAN, such as steering wheel data, due to the encoding of the information on the data bus.

Therefore, there is an interest in classifying or mapping the behavior of drivers, which is the subject of several studies. In [18], it was proposed that driving behaviors can be divided into four categories: cautious, average, experienced, and reckless. Frequent use of acceleration and braking was used as parameters to assess how close the steering is to the control limit. Another approach was presented by [19], classifying driving as cautious, normal, and aggressive through temporary evaluations based on the transients of the actions. In [20], a system was developed to evaluate a driver’s performance and attitude based on their maneuvers, including stable tracking, acceleration, approach, braking, and opening. The maneuvers were divided into two groups: maneuvers performed safely, and maneuvers performed dangerously [20]. Based on these classifications, the driver’s behavior can be divided into two more precise classes: a “normal/calm/cautious attitude” behavior and a “fast/hasty/fewer cautions attitude” behavior. If driver behavior can be classified, devices for that purpose can assist Intelligent Driver Advice Systems [21,22]. In addition, monitoring the driver’s behavior can impact the reduction of actions, with about one third being related to decision errors [23] such as driving at high speed, carelessness, tiredness, and impatience, among others [24].

Regarding the recognition of driving patterns and/or driver identification, the literature presents works such as [25], which aimed to recognize safe and unsafe driver behaviors using machine learning models [26,27]. This work used 51 signals, focusing on vehicle speed, engine speed, engine load, throttle position, steering wheel angle, and brake pedal pressure from the CAN bus, based on ten drivers and resulting in accuracy classifiers above 90%. Another study, ref. [28], aimed to identify the driver using only CAN bus signals (12 signals, including steering wheel angle, velocity, acceleration, vehicle velocity and heading, engine RPM, gas pedal position, brake pedal position, forward acceleration, lateral acceleration, torque, and throttle position). It used 10 cars and 64 drivers, and the proposed system obtained an accuracy of 76% between two drivers and 50% for five drivers. In both cases, the authors did not collect signals from the driver and did not conduct correlations between the driver’s and car’s signals. Moreover, recently, ref. [29] presented a study of driving behavior identification by deep learning, with automatic activity recognition on real-time-based CAN bus data and without feature selection, based on 51 features from CAN bus sensor data. The proposed model reached above 95% without feature extraction and did not correlate with sensors without CAN bus.

Several other works have used inertial sensors to identify the patterns of both the direction and the driver. Among them, refs. [30,31,32,33,34,35] used the camera and sensors on a smartphone (accelerometer, gyroscope, magnetometer, and GPS) for their purposes. Table 1 present a technical summary of these articles, mainly pointing out the results and limitations of each one.

Thus, some questions arise to be answered:Could the driver’s leg movement information (electromyography and inertial modules) provide studies of driving patterns?Could this driver’s information be somehow correlated with data from the vehicle’s internal sensors?Could these data jointly be used to feed computational intelligence tools to assess driver behavior?Since there is a correlation between driver data and vehicle data, would data from the vehicle alone be sufficient to assess driver behavior? And how accurately?What would be the minimum set of sensors inside the vehicle that could provide enough information to correctly describe the driver’s driving mode and still identify him? And how accurately?

From this perspective, this work identifies the driver’s driving profile using machine learning techniques. The data are acquired from information on the vehicle’s CAN bus and through sensor fusion, which was instrumented in the vehicle (brake pedal) and driver (surface electromyography and inertial sensors). In addition, the feasibility of identifying the driver from the data collected from their driving is presented. One of the differentials of this work is the use of sensor signals in the brake pedal. There are few examples in the literature verifying whether they are necessary for the classification process. In this paper, we explore this signal’s influence to classify both the driver’s identity and individual behavior.

The rest of this paper is divided as follows: Section 2 presents the Materials and Methods. First, it presents the details about the methodology, approaching the experimental protocol, the car and driver instrumentation and the process of data acquisition. Then, the methodology of signal processing and pattern recognition processing are presented. In Section 3, the results and their discussion are presented. Finally, in Section 5 the conclusions are discussed.

## 2. Materials and Methods

This section presents the materials and experimental protocol, which are summarized in Figure 1. First, an experimental protocol was developed to define which conditions and actions the drivers would perform. After defining the behavior patterns and the path to be developed, the vehicle and driver were instrumented, and the data acquisition step was started. The obtained data were processed and classified using machine learning techniques to distinguish the driver and their behavior. The experiment has the approval of the Ethical Committee for Research on Humans of the Federal University of Technology—Paraná (CAAE: 50086021.8.0000.5547).

### 2.1. Experimental Protocol

Before starting the data acquisition, it was necessary to verify the conditions related to the driving patterns. In this experiment, two driving patterns were chosen: a normal, calm state with cautious driving and a more abrupt, hurried, and less cautious condition. These two classes are categorized into specific actions in Table 2. These inductions were determined based on the drivers’ way of driving. For example, acceleration and steering can be slow and gradual (representing the calm and more cautious state) or abrupt and emphatic (representing haste and a less careful state). This less attentive action may indicate a desire or need to overtake, being driven by anxiety, or a rush to get to an appointment for which the driver is late. Similarly, abrupt braking can represent haste or the emergence of unforeseen events, such as a vehicle ahead slowing down or the appearance of a sudden obstacle (such as a pothole in the road, an unmarked speed bump, and an animal crossing the road).

The actions mentioned in Table 2 allow the creation of different situations for each participant to differ between basic steering car behaviors and pedal actions (braking and acceleration). Drivers were instructed to perform the fast route in approximately 70% of the time used for the slower route. Additionally, steering wheel steering and braking maneuvers were equally different for each case (alternating between abrupt and slow). These actions bring a dose of subjectivity to the data set, as each driver, although guided, acted differently, according to personal interpretation, bringing differences between their slow and fast perspectives and having as the only single parameter the maximum times for execution of the routes. The route taken by the drivers is shown in Figure 2, which is a simple test of situations and actions present in traffic, such as curves and brakes.

### 2.2. Instrumentation and Data Acquisition

After developing the experimental protocol, the data acquisition started, beginning the instrumentation for both driver and vehicle. Figure 3 presents the used sensors and the places where they were placed. First, a device to monitor the CAN bus was used. It is denominated in this work as Automotive Data Acquisition Center (ADAC), which is connected by the ODB2 interface. This device can acquire three parameters with a 1000 samples/second sampling rate, with local storage and wireless data transmission. The analog–digital converters have 12-bit resolution with a range from 0 to 3.3 V. More details of technical aspects are available on [9]. The vehicle acquired instantaneous velocity, rotations by minute, and analog accelerator position (gas pedal). The scan of CAN parameters occurred with a 10 samples/second rate. This device also provides the acquisition of external signals, which were allocated to instrument the brake pedal, the foot, and the driver’s leg.

A sensor with linear displacement was placed on the brake pedal and connected to one ADAC analog input. The need for this sensor is because only the information about the drive of the brake pedal is available on the CAN bus. The signal is binary, which notifies if the pedal is or is not pressed. The used sensor consisted of a slider potentiometer with a course of 70 mm, fixed on the vehicle panel, and a metal rod connected to the brake pedal. The circuit had a voltage regulator of 3.3 V (AMS1117) to provide the same input voltage for the sensor.

The second analog input of ADAC was connected to the surface electromyography (sEMG) circuit. The circuit had a gain of 54 dB (provided by Instrumentation Amplifier INA128), second-order Butterworth filters to delimit the frequency band of 10 to 500 Hz, and offset of 1.7 V to conform to the analog–digital converter range (0 to 3.3 V). As the sEMG circuit is bipolar, the electrodes were placed on the driver’s right leg on the tibialis anterior muscle, which relates more to the pedal activation movement. The reference electrode was placed above the knee. As the sEMG circuit and sensor on the brake pedal were connected to the ADAC analog input, both had an acquisition of 1000 samples/second.

The last sensor used was an inertial measurement unit (IMU) placed on the driver’s right foot using Velcro straps. An MPU6050 module combines a three-axis accelerometer and a three-axis gyroscope, communicating with the ADAC via the Bluetooth protocol. This sensor monitors the foot’s movement while driving the vehicle. This sensor had an acquisition rate of 50 samples per second.

Data from five drivers were acquired (four males and one female, aged 30 ± 6.5 years old). For each volunteer, the data acquisition started after placing the sensors on the driver and in the car. Drivers were instructed to drive the vehicle considering actions that proposed different driving behaviors, targeting the driving mode in both classes, as shown in Table 2. The volunteers drove six laps of the route in Figure 2, three in cautious conditions and three in less careful conditions. Each condition was acquired at different times: 250 s for cautious and 200 s for less cautious.

### 2.3. Signal Processing and Pattern Recognition

The signal processing steps are presented in Figure 4. The acquired data were saved on text files, and these files were uploaded onto Matlab ^®^ scripts. Samples of sEMG with twice the standard deviation were discarded to reduce the noise from sEMG signals. They are more related to interferences and were replaced by the previous values of the signals. Thus, the first step in signal processing was to normalize all sensor data using the maximum value of each sensor. The range of attributes was maintained from 0 to 1.

Data segmentation (second step) was performed based on sEMG signals. The double onset method was applied to identify sEMG activation. When the sEMG signal was identified, there was foot movement, and at this time, the data should be segmented (CAN bus, brake pedal, and sEMG information). The moving average filter smoothed the sEMG signals with n-samples. A value of 60% of the mean for all distributions was used as a threshold. Moreover, it was verified if this value was maintained for 1 s to consider an sEMG activation. The values that did not belong to conditions of sEMG activation were turned to zero. Thus, all of the sensor data were segmented in fixed windows of 2 s for all sensors, following each sensor’s sampling rate.

The next step was feature extraction, performed for each sensor and parameter. Table 3 presents the extracted features for each sensor. The Mean Absolute Value (MAV) is a feature widely used in sEMG processing [36,37]; in the inertial sensors, the mode was extracted for each axis. For the data obtained from the CAN bus and brake pedal, we extracted the static parameters, such as mode, mean, median, and maximum values.

The analysis of the feature set is organized to verify the influences of features provided by the driver and the vehicle in order to increase their complexity. By contrast (at the same time), we verified the influence of the brake pedal, which was instrumented especially for verifying its effect on the classification process. The features were organized in feature sets after the feature extraction. The used combinations are presented in Table 4, which presents their respective number of tests and the data origin.

In the pattern recognition step, this work applied three classifiers: k-nearest neighbors (k-NN), support vector machine (SVM), and random forests. The k-NN classifier was programmed with the five nearest neighbors. In SVM, a kernel quadratic polynomial function was chosen due to the non-linear nature of the data. Random forests were programmed in bagged tree manner (restricting 30 trees and 189 nodes by tree). The classifiers were evaluated by accuracy (how much the classes are correctly classified) and specificity (influence of false positives during the pattern recognition process) metrics. The k-fold cross-validation was performed with ten folds for all datasets.

Before classification, the data were labeled in two approaches. The first label was related to driver behavior in two classes (cautious and less cautious) for all datasets, without identification of separation by the driver. The second approach consisted in classifying first the identity of the driver and, then, their behavior (calm and aggressive directions).

To verify the different classifiers’ performance and the different sets of attributes, Friedman’s statistical test was used and, later, Tukey’s post-hoc test. The Friedman statistical test, an extension of the Wilcoxon statistical test, is a method used to assign ranks to observations to verify whether samples belong to the same observation. Tukey’s post-hoc test made it possible to prove the equivalences between the groups.

## 3. Results and Discussion

To exemplify the signals obtained from the sensors on the vehicle and the driver, Figure 5 presents the data acquired from ADAC for the less cautious state. Figure 5 shows the data obtained from the CAN bus (velocity, RPM, and position) and the signals from the sEMG circuit and potentiometer placed on the brake pedal. The axes of the CAN bus are normalized because each one has its unity. Maximum values were used to perform the normalization. The signals were synchronized due to the presence of different sampling rates. Additionally, the three axes (x, y, and z) are presented for the gyroscope and accelerometer sensors.

First, one can note that there are regions where there are sEMG activations when the brake pedal passes from the press to rest states. The tibialis muscle (where the sEMG circuit was placed) was activated when drivers removed their feet from the brake pedal. It is related to the paths where the drivers should reduce their velocity. This can be seen in Figure 5a. The acquired variables present the change of the accelerator and brake pedals, as highlighted in Figure 5b,c. Whenever the brake pedal is pressed, the sEMG activation is detected before. Straight paths on the road denote the moderate use of the brake pedal, and the sEMG and accelerator signals are more visible (Figure 5d). The end of the route is exemplified in (Figure 5e), where velocity is zero; the RPM, sEMG signal, and accelerator pedal sensor are at minimum, and the brake pedal is at the maximum value. Concerning the inertial sensors, one can note that each change of interaction between the accelerator and braking pedals (made by the foot, where the IMU is placed) is related to the signals on the accelerometer and gyroscope. As they were fast movements, the inertial sensors acquired the signals as spikes.

Comparing the two conditions (I and II) of Figure 5, one can note that there is some relation between the driver behaviors. On the IMU sensors, less caution presented values with peaks higher than cautious driving, which can be observed on the sEMG signal and in the interaction of the brake pedal sensor. In addition, the data obtained from the CAN bus have some differences for the two situations, which could indicate differences between these two conditions and identify the driver.

Based on the activation of sEMG, the signals were segmented. It resulted in 709 instances for the 5 drivers. Thus, the features presented in Table 3 were extracted for each sample and normalized in values between 0 and 1. Figure 6 shows the distribution for each attribute for the two approaches.

Starting with driver behavior, one can note that some features have deviations from their median, even present instances with similar values on the distribution. It was verified for parameters extracted from the CAN bus, such as velocity, RPM, and accelerator pedal attributes. In addition, cautious behavior (C) presented more caudal distribution than less caution (L). One can note that gyroscope data show a high quantity of outliers with concentrated distributions. On the other hand, the classes are not easily separated for the individual’s identification. The distributions are similar, and they have a high number of outliers for all features. This exemplifies the complexity of the classification problem due to the elevated level of similarity among the classes.

Based on the fourteen different feature sets presented in Table 4, Figure 7 presents the parameters of accuracy and sensibility qualitatively using column charts to recognize the drivers’ behavior and their identity, respectively, based on k-NN, SVM and RF classifiers. Quantitatively, Table 5 presents the best results for each feature set for behavior identification and driver identification. Bold results mean the highest values for both conditions. Regarding the classifiers, it can be seen that the random forests presented the best results with the combination of features with a large number of sensors. The highest accuracy occurred when all sensors were applied to the classifiers (sets 13 and 14), both to recognize the drivers’ behavior and identity.

In attention to Figure 7I, we can verify that the 14th set tested presented the highest accuracy (0.94) for driver’s behavior due to containing data from the vehicle and the driver; sets containing only data from the drive (10, 11 and 12), presented the highest accuracy values around 0.91; however, sets 5, 7, 8, and 9 represented a decrease around of 10% of accuracy (comparing with set 12). The sensibilities remained close to the accuracies, indicating a low number of false positives. The sets composed by RPM, acceleration pedal, or the velocity sensor (sets 1 to 5) obtained better accuracy when combined with the brake pedal (sets 7 to 9). The use of the brake as an isolated characteristic (set 6) was not enough to provide high precision by itself. Still, it contributed to the other sets (groups 7, 8, and 9) recognizing the drivers’ behavior.

Regarding drivers’ identity, Figure 7II shows higher accuracies for some combinations of features and classifiers. However, the highest accuracy obtained was 0.93, using all vehicle features and the bagged classifier. One can note that the drivers’ identity managed to be recognized even with high specificity. In some cases, false positives were lower than the identification itself; that is, the system can find a driver identification profile. It is noteworthy that the brake pedal instrumentation remains a highly relevant factor since the classifiers showed similar distributions between the use of all sensors and the data only from the brake pedal instrumentation. An interesting case is that using the accelerator with the brake also has a lower error rate but low sensitivity.

Quantitatively, Table 5 presents the best results for each feature set for behavior and driver identification. Bold results mean the highest values for both conditions.

As shown in Table 5, when all available attributes (vehicle and driver, at feature sets 13 and 14) were used, an accuracy greater than 0.93 was obtained in the behavior identification and 0.96 in the identification of the driver. Using only the in-vehicle sensors added to the brake pedal, the accuracy was greater than 0.80 for identifying the behavior (showed in lines of feature sets 8 and 9), and approximately 12% lower accuracy was obtained using vehicle data only. Figure 8 presents the statistical analysis used to analyze the correlation of the data.

The Tukey post-hoc from Friedman statistical test was used to evaluate the distribution of results with a confidence interval of 95% (*p*-value < 0.05) in two perspectives. For the driver’s behavior (Figure 8I) and for the driver’s identification (Figure 8II). For each perspective, the analysis for the classifiers is presented (two sub-figures on the left side, indicated by letter (a)) and the analysis for each sensor (two sub-figures on the right side, indicated by letter (b)). The distributions presented in these results show a good correlation between vehicular data and data obtained from the driver, suggesting that further studies may be promising to improve the accuracy of rates based exclusively on vehicle characteristics for behavior identification and driver identification.

### Discussion about the State-of-the-Art

In summary, the results obtained indicate:
(a)Using only vehicle data: accuracy of 0.82 and 0.66 from behavior and driver identification, respectively;(b)Only driver data: accuracy of 0.91 and 0.96 from behavior and driver identification, respectively;(c)Vehicle and driver data: (accuracy of 0.94 and 0.96) from behavior and driver identification, respectively.

Evaluating the Tukey post-hoc from the Friedman statistical test, it could be concluded that there is a good correlation between the data coming from the vehicle’s sensors and the data coming from the driver’s sensors. This conclusion allows us to continue with more in-depth future work on pattern recognition, considering only vehicle sensor signals for driving profile assessment or driver identification.

When comparing our results with the state-of-the-art results, as summarized in Table 6, we noticed that no work analyzed this correlation; they just assumed that pattern recognition studies could be performed. As for the results obtained, our accuracy rates were satisfactory when compared to other studies and when considering the low number of parameters used. The data we analyzed were collected from a vehicle with no available data on brake pedal excursion, which prompted us to instrument the brake pedal and externally add this signal. Likewise, steering wheel steering data were not collected from CAN due to difficulties in decoding identifiers and data in the vehicle used. However, it is understood that these data are crucial for improving accuracy rates (future work will be devoted to deepening pattern recognition procedures with better datasets). However, they were not essential for the study of the correlation of these signals.

## 4. Practical Implications and Limitation of the Study

The practical implications of this study point to future work, which will allow embedded systems to generate signals (light or sound) in real-time in order to indicate to the driver that his driving pattern may be aggressive and, therefore, susceptible to accidents. In this perspective, driving assistance systems can even generate reports of the driving mode so that logistics companies can monitor the performance of their drivers.

Still, in this perspective, it is understood that this study can be expanded by considering a larger number of vehicular information and evaluating the relevance of vehicular sensors for this purpose. Still, improvements in the experimental methodology and an increase in the number of factors to be analyzed and the number of drivers can significantly improve the performance of pattern recognition tools.

It is important to point out that the focus of this work was to verify the correlations of biomedical (electromyography) and biometric (inertial) sensors with some vehicle dynamics sensors. After this correlation, a preliminary analysis of behavior pattern recognition and driver identification was carried out. Future work will aim at the insertion of new sensors and their evaluation, as well as a more thorough evaluation of different artificial intelligence algorithms for these purposes, to improve accuracy.

## 5. Conclusions

The growing investment in the field of sensors and real-time monitoring systems has enabled the development and advancement of different security systems, such as anti-lock braking systems (ABS), electronic stability control (ESC), traction control system (TCS), and advanced driver assistance systems (ADAS). In this line, vehicles increasingly provide sensors that can use their data for different purposes.

In this line, this work proposed the evaluation of a set of sensors with data available through the CAN network, together with sensors additionally installed in the vehicle and sensors installed in the driver, to analyze the recognition of driving behavior patterns and also of driver identification. Speed, RPM, and accelerator pedal position data from CAN were chosen; instrumented brake pedal data and data from inertial sensors positioned on the driver’s foot and from leg electromyography.

The objective was to analyze the hypothesis of recognition of behavior patterns and driver identification through vehicle data correlated with driver data. An experimental protocol was developed to label two forms of driving (calm and less calm). Data from five conductors were collected. After pre-processing the data and extracting the features, k-NN, SVM, and RF classifiers were applied. The results show an accuracy greater than 93% for identifying driving behavior when correlating vehicle data with driver data. Approximately 12% lower accuracy was obtained using vehicle data only. It was noted that using the brake pedal position sensor improved the accuracy rate.

The results show a good correlation between vehicular data and data obtained from the driver, suggesting that further studies may be promising to improve accuracy rates based exclusively on vehicle features, both for behavior identification and for identification of the driver.

Thus, once the correlation of the driver’s sensor data is presented, with vehicle data, future works will include the exclusion of the driver’s instrumentation and the insertion of a larger set of sensor data exclusively from the vehicle. In addition, the increase in the number of drivers and the increase in the perimeter of the route, with the insertion of new events, may contribute to improving the generalization of pattern recognition tools. Still in the line of data processing, further studies of feature extraction and windowing may improve the accuracy rates obtained by the classifiers. Finally, we want to carry out the procedures to embed the data processing in hardware for real-time analysis and generation of driver guidance signals.

## Figures and Tables

**Figure 1 sensors-23-00263-f001:**
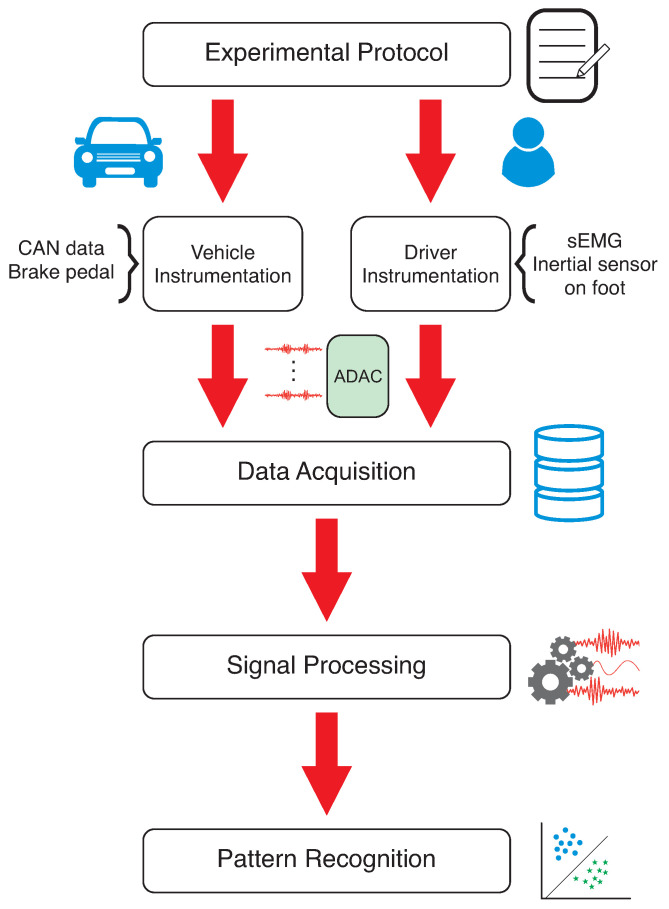
Methodological procedure flowchart.

**Figure 2 sensors-23-00263-f002:**
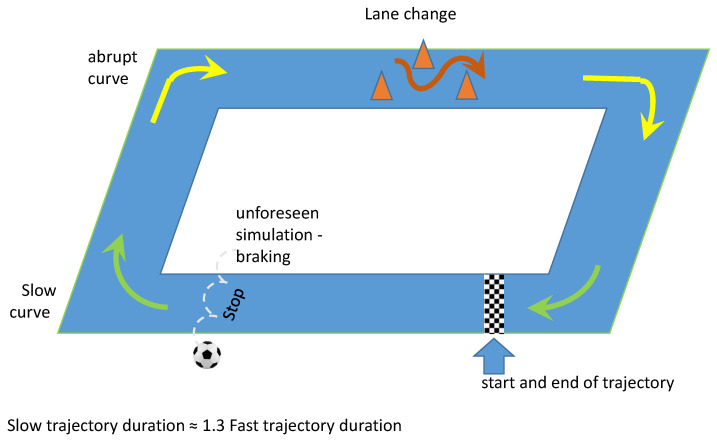
Route applied in the tests. The drivers started and ended the route at the same place.

**Figure 3 sensors-23-00263-f003:**
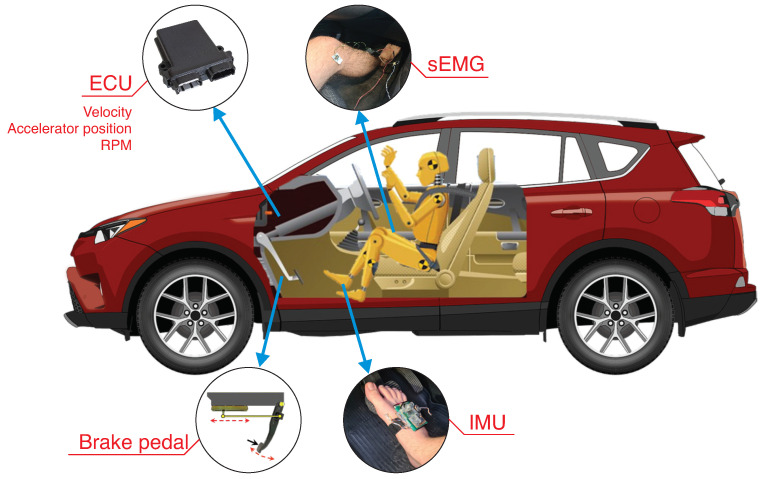
Data source (vehicle sensor information from the CAN network and vehicle instrumented sensors).

**Figure 4 sensors-23-00263-f004:**
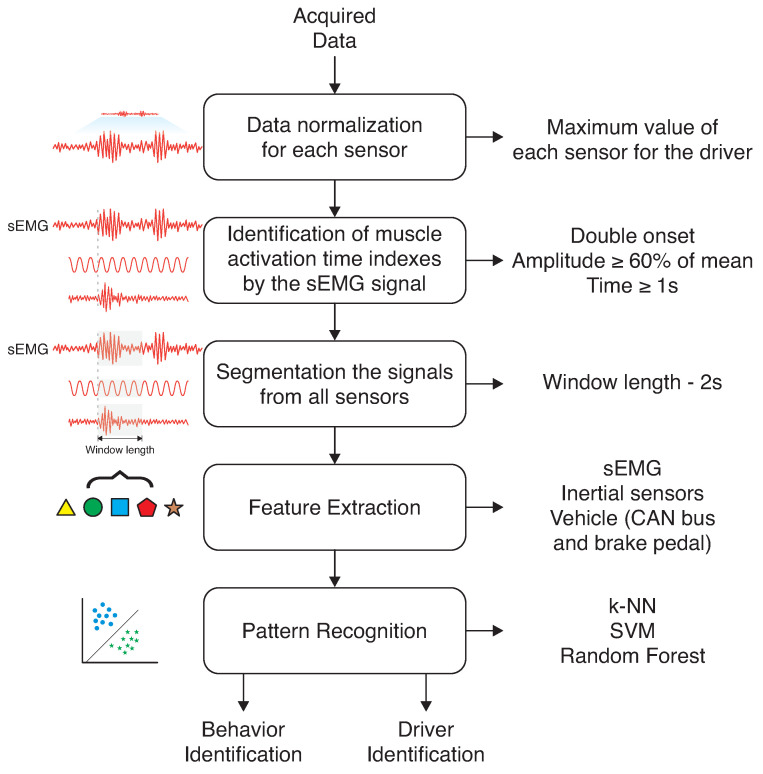
Signal processing steps.

**Figure 5 sensors-23-00263-f005:**
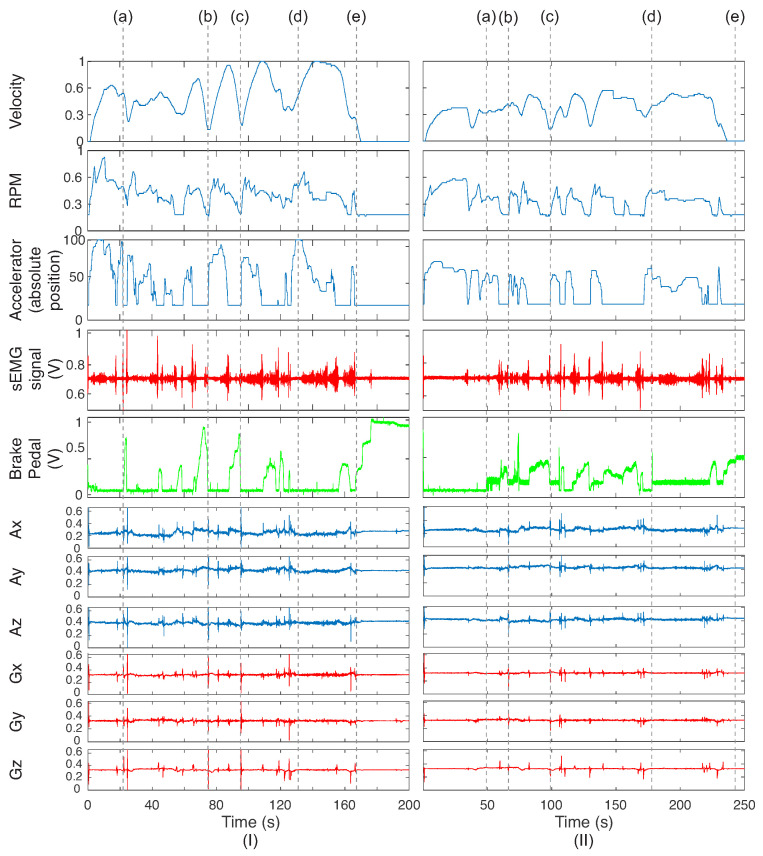
Example of signals obtained from the vehicle and driver for each sensor on (**I**) less cautious and (**II**) cautious state: CAN bus (accelerator, RPM, velocity data), sEMG signals, brake pedal sensor, and IMU data (accelerometer and gyroscope for the three axes). The values of the CAN bus are normalized for the acquisition. Some moments are highlighted for some conditions, such as step on brake pedal (a), pressing the accelerator (b), use of brake on the road (b,c), straight paths on the road (d), and end of the route (e).

**Figure 6 sensors-23-00263-f006:**
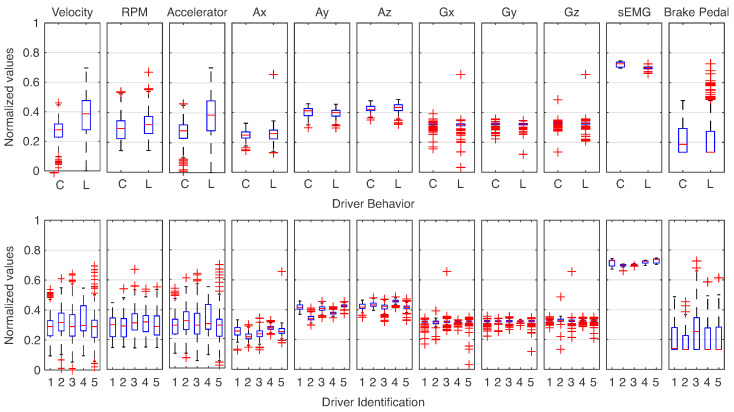
Distributions for each class based on the features extracted from Table 3. The data were separated for driver behavior for Cautious (C) and Less Cautious (L) conditions. The features of five drivers were also explored. All of the features were normalized between 0 and 1.

**Figure 7 sensors-23-00263-f007:**
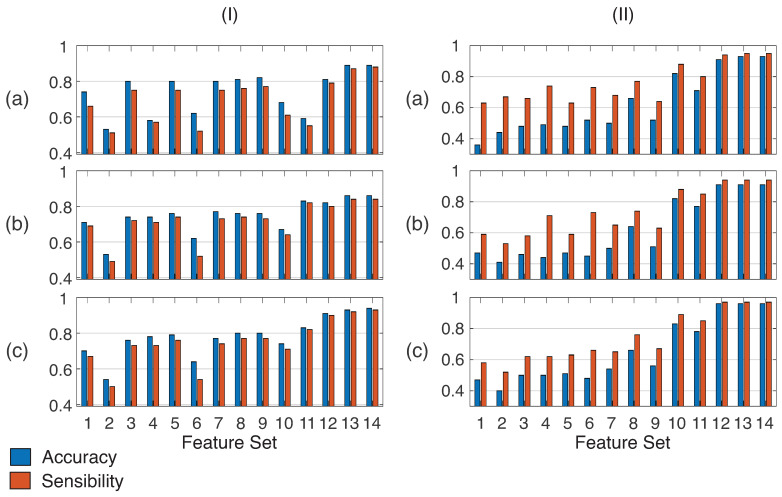
The accuracies and sensibilities obtained from each feature set for (**I**) drivers’ behavior and (**II**) their identity in the following classifiers: (**a**) k-NN, (**b**) SVM, and (**c**) RF.

**Figure 8 sensors-23-00263-f008:**
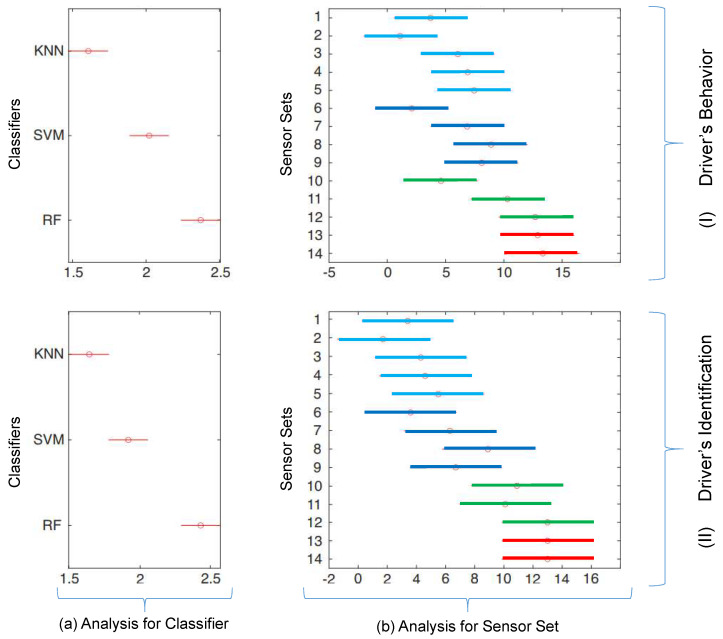
Distributions for the Tukey post-hoc from Friedman statistical test to evaluate the distribution of results with a confidence interval of 95% (*p*-value < 0.05) for (**I**) driver’s behavior and (**II**) driver’s identification. (**a**) presents the analysis for the classifiers and (**b**) the analysis for each sensor set.

**Table 1 sensors-23-00263-t001:** Overview of the state-of-the-art.

REF	Objective	Sensor/Data	Main Results	Study Limitation
[25]	Recognize safe and unsafe driver behaviors using machine learning models	Data from CAN bus (51 signals, focus on vehicle speed, engine speed, engine load, throttle position, steering wheel angle, brake pedal pressure)/10 drivers	Classifiers presented accuracies above 90	Signals from the driver were not considered
[28]	Identify the driver using a shorter segment of road	CAN bus signals (12 signals, including steering wheel angle, velocity, and acceleration; vehicle velocity and heading, engine RPM, gas pedal position, brake pedal position, forward acceleration, lateral acceleration, torque, throttle position)/10 cars and 64 drivers	System obtained accuracy of 76% between 2 drivers and 50% for 5 drivers	Signals from the driver were not inserted on the analysis.
[29]	Driving behavior identification by deep learning, with automatic activity recognition on real-time-based CAN bus data and without feature selection	51 features from CAN bus sensor data/Use an Ocslab driving dataset, with 94401 records, ten drivers, two rounds trips	The proposed model reached above 95% without feature extraction	Used a database and did not correlate with sensors without CAN bus.
[30]	Detect abnormal behavior (such as speed changes and steering) during driving using a smartphone	Accelerometer and gyroscope from smartphone/4 smartphones in a taxi	Four typical dangerous driving behaviors were detected: abnormal speeding up or down, steering, weaving, and operating smartphone, with 90% of accuracy. A novel method to compute the yaw angle was proposed.	Use only smartphone to acquire the data, without correlation with variable on a car. Difficulty in weaving detection.
[31]	Examine driver behavior using smartphone sensors by harsh accelerations and harsh decelerations across junctions and road segments	Sensors on smartphone (accelerometer, gyroscope, magnetometer, and GPS) and traffic characteristics by inductive loops/data from 303 drivers in two urban expressways	Authors realized that there were harsh increases on road segments if average traffic flow per lane also increases	Authors did not recognize patterns, they used prediction models.
[32]	Recognize driver behavior (normal, aggressive, and drowsy driving) using an LSTM and sensors on a smartphone	Accelerometer (x, y, and z acceleration), gyroscope (roll, pitch, and yaw angles), GPS (vehicle speed), and camera sensor (distance ahead and number of detected vehicles)/UAH-DriveSet (two types of road, six drivers and vehicles, and three driving behaviors)	LSTM provides a mean f1-score of 0.91 for all sets without feature extraction	The authors used a database and did not explore the influence of data obtained from a car.
[33]	Recognize driving events using DTW and KNN using a smartphone, such as right and left turns, right of left lane change and road anomalies. The movements were analyzed along the three axes. Drivers were classified with abnormal (aggressive) and normal driving behavior.	Accelerometer, gyroscope, and GPS from smartphone on the center of vehicle dashboard/two drivers in two different vehicles (540 driving events)	About 99% for driving detection using KNN and about 97% for driving behavior with DTW	Authors only used smartphone data, without correlation with variables on a car.
[34]	Two approaches to recognize driver behavior (normal, drowsy, and aggressive, and aggressive and non-aggressive) using an LSTM model.	Accelerometer, gyroscope, GPS, preprocessed vehicle detection (video recordings)/UAH-DriveSet (two types of road, six drivers and vehicles, and three driving behaviors)	Winner model with f1-score of 99%	The authors used a database.
[35]	Detection of dangerous states for accident prevention by driver behavior monitoring.	Smartphone (accelerometer, gyroscope, GPS, and microphone) and camera/Data acquired from 10 volunteers	States such as distraction and drowsiness were recognize	Authors did not provide overall accuracy and data from the bus were not applied.

**Table 2 sensors-23-00263-t002:** Driver behavior actions based on distinct actions for classification between cautious (calm, or slower) and less cautious (hurried, more abrupt).

Action	Driving Type	
	Calm, Cautious	Hurried, Less Cautious
Initial acceleration	Slow, gradual	Abrupt, fast
Breaking	Slow, gradual	Abrupt
Unforeseen, simulation	Slow braking	Abrupt braking
Regaining steering control	Slow, gradual	Abrupt, fast
Lane change	Slow, gradual	Abrupt, steering wheel movement
Curve	Moderate speed, reduction	Minimum speed, reduction

**Table 3 sensors-23-00263-t003:** Sensor data origin and extracted features.

Sensor	Features
sEMG	Mean absolute value (MAV)
Accelerometer	Mode in each axis (X, Y, Z)
Gyroscope	Mode in each axis (X, Y, Z)
Break	Median
Velocity	Mean
RPM	Mean
Accelerator	Median

**Table 4 sensors-23-00263-t004:** Sets of attributes selected for evaluation of classifiers. Sets 1–5 are based on only features extracted obtained from in-vehicle sensors; sets 6–9 are based on insertion of the break pedal features; sets 10–12 are based on only features extracted from the driver; and the last sets (13–14) are sets with vehicle and drive features.

Sensor Set	Data Origin	Selected Features
1	Vehicle	Velocity
2	Vehicle	RPM
3	Vehicle	Velocity + RPM
4	Vehicle	Accelerator
5	Vehicle	Velocity + RPM + Accelerator
6	Vehicle	Brake Pedal
7	Vehicle	Accelerator + Brake Pedal
8	Vehicle	Velocity + RPM + Brake Pedal
9	Vehicle	Velocity + RPM + Accelerator + Brake Pedal
10	Driver	Inertial
11	Driver	sEMG
12	Driver	Inertial + sEMG
13	Vehicle + Driver	Velocity + RPM + Accelerator + Inertial + sEMG
14	Vehicle + Driver	Velocity + RPM + Accelerator + Brake Pedal + Inertial + sEMG

**Table 5 sensors-23-00263-t005:** Best accuracies obtained in the best feature sets in the two approaches.

		Behavior Identification			Driver Identification		
**Feature Set**		**k-NN**	**SVM**	**RF**	**k-NN**	**SVM**	**RF**
In-Vehicle Sensor Data only	5	0.80	0.76	0.79			
	8	0.81	0.76	0.80	0.66	0.64	0.66
	9	0.82	0.76	0.80	0.52	0.51	0.56
Driver Data only	12	0.81	0.82	0.91	0.91	0.91	**0.96**
Vehicle and Driver Data	13	0.89	0.86	0.93	0.93	0.91	**0.96**
	14	0.89	0.86	**0.94**	0.93	0.91	**0.96**

**Table 6 sensors-23-00263-t006:** Comparison of the results with the state-of-the-art.

REF	Objective	Sensor/Data	Main Results	Study Limitation
[25]	Recognize safe and unsafe driver behaviors using machine learning models	Data from CAN bus (51 signals, focus on vehicle speed, engine speed, engine load, throttle position, steering wheel angle, brake pedal pressure)/10 drivers, resulting in 26 h	Classifiers presented accuracies above 90	Signals from the driver were not considered
[28]	Identify the driver using a shorter segment of road	CAN bus signals (12 signals, including steering wheel angle, velocity, and acceleration; vehicle velocity and heading, engine RPM, gas pedal position, brake pedal position, forward acceleration, lateral acceleration, torque, throttle position)/10 cars and 64 drivers	System obtained accuracy of 76% between two drivers and 50 for five drivers	Signals from the driver were not inserted on the analysis.
[29]	Driving behavior identification by deep learning, with automatic activity recognition on real-time-based CAN bus data and without feature selection	51 features from CAN bus sensor data/Used a Ocslab driving dataset, with 94401 records, ten drivers, two rounds trips	The proposed model reached above 95% without feature extraction	Used a database and did not correlate with sensors without CAN bus.
This work	Analyze if there is a correlation between signals from the vehicle (can) with data instrumented on the driver’s leg/foot to propose a system for the detection of dangerous states for accident prevention by driver behavior monitoring, based on two standards of driver behaviors and identifying of the driver.	An acquisition board developed by the authors was used, and only three CAN data were captured (velocity + RPM + accelerator), in addition to externally instrumented data (break pedal) and driver leg/foot movement signals (sEMG and inertial movements)	The results presented a good correlation between driver and vehicle signals. About the accuracies: (a) Using only vehicle data: accuracy of 0.82 and 0.66 from behavior and driver identification, respectively; (b) Only drive data: accuracy of 0.91 and 0.96 from behavior and driver identification, respectively; (c) Vehicle and driver data: accuracy of 0.94 and 0.96 from behavior and driver identification, respectively.	Number of users, a limited number of parameters, but enough to analyze and prove the correlation between driver and vehicle data, for future study without driver data.

## Data Availability

Not applicable.

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
