# Peer review of "Correlation Analysis of In-Vehicle Sensors Data and Driver Signals in Identifying Driving and Driver Behaviors"

_sensors, 2022, doi:10.3390/s23010263_

Round 1
Reviewer 1 Report
The objective was to analyze the hypothesis of recognition of behavior patterns and driver identification through vehicle data correlated with driver data. An experimental protocol was developed to label two forms of driving (calm and less calm). Data from 5 conductors were collected. After pre-processing the data and extracting the features, k-NN, SVM, and RF classifiers were applied. The results showed an accuracy greater than 93% for identifying driving behavior when correlating vehicle data with driver data. Approximately 12% lower accuracy was obtained using vehicle data only. It was noted that using the brake pedal position sensor improved the accuracy rate. The results show a good correlation between vehicular data and data obtained from the driver, suggesting that further studies may be promising to improve accuracy rates based exclusively on vehicle features, both for behavior identification and for identification of the driver.
Author Response
Dear reviewer, we thank you for carefully evaluating our manuscript. We hope
that, together with the other reviewers’ comments, we have further improved the presentation of this work.
Reviewer 2 Report
1. The research novelty is not clear. The authors can make a section on 'Research motivation and contributions' at the end of introduction to clearly highlight what novel this study adds to the existing literature.
2. The authors can present a literature summary where they can highlight the key findings in one column and key limitations in another column - which will help in identifying the research gaps and will be easy to comprehend by the readers.
3. In Table 1, the first cell is empty. Please check.
4. Please check that all the abbreviations are explained at least once at their first use in the manuscript.
5. The authors can separate 'results and discussion' into two individual sections on 'results' and 'discussion'. In the results section, they key findings and interpretations can be summarized, and in the discussion section, the discussion of obtained findings in light of past literature needs to be provided. At present, the discussion with respect to similarities and contrasts with previous studies is missing.
6. The final conclusions can be listed out in bullets for easy understanding of the readers.
7. No limitations of the study are provided in the paper. The authors need to make a section on 'Limitations and future research scope' to highlight the limitations observed in the study and how the future researchers can take forward this research.
8. In the end, a section on 'Practical implications' is required where the authors can throw some light on the practical use of their findings and takeaways from policy makers and transport officials.
9. In the abstract, the authors can add a sentence on research motivation in the beginning, and practical applications in the end, which will give a wholesome touch to the abstract.
10. Instead of using terms like 'accidents', the authors need to use 'crashes' as collisions do not always happen accidently, and crash/crashes is a better accepted term in the international literature and road safety organizations.
Author Response
Comments and Suggestions for Authors
1. The research novelty is not clear. The authors can make a section on ’Research motivation and contributions’ at the end of introduction to clearly highlight what novel this study adds to the existing literature.
A - Dear reviewer, we are very grateful for your comments to improve the quality of the presentation of our work. Regarding this topic, we adjusted the text in the introduction to improve the state of the art and highlight the contributions/hypotheses of the work.
2. The authors can present a literature summary where they can highlight the key findings in one column and key limitations in another column - which will help in identifying the research gaps and will be easy to comprehend by the readers.
A- Dear reviewer, we very much appreciate your observation. Regarding this topic, we adjusted the state of the art and inserted a table with results and limitations of the state of the art works.
3. In Table 1, the first cell is empty. Please check.
A - Ok, fixed.
4. Please check that all the abbreviations are explained at least once at
their first use in the manuscript.
A - Ok, fixed.
5. The authors can separate ’results and discussion’ into two individual sections on ’results’ and ’discussion’. In the results section, they key findings and interpretations can be summarized, and in the discussion section, the discussion of obtained findings in light of past literature needs to be provided. At present, the discussion with respect to similarities and contrasts with previous studies is missing.
A - Dear reviewer, thank you for your suggestion. To try to answer the three reviewers’ questions, we preferred to leave a section of results and discussions together so that the results could be discussed sequentially in blocks. Additionally, we discuss the main results obtained with state of the art.
6. The final conclusions can be listed out in bullets for easy understanding of the readers.
A - Dear reviewer, thank you for your suggestion. However, to meet the expectations of other reviewers, we consider that the visibility of the results achieved can be maintained continuously and not in bullets at the conclusion. Note that this has been improved at other points in the article, such as at the end of the results, including the addition of sections on Practical implications and Limitation of the study.
7. No limitations of the study are provided in the paper. The authors need to make a section on ’Limitations and future research scope’ to highlight the limitations observed in the study and how the future researchers can take forward this research.
A - Dear reviewer, we very much appreciate your observation. We added sections with this information in the text.
8. In the end, a section on ’Practical implications’ is required where the authors can throw some light on the practical use of their findings and takeaways from policy makers and transport officials.
A - Dear reviewer, we very much appreciate your observation. We added sections with this information in the text.
9. In the abstract, the authors can add a sentence on research motivation at the beginning, and practical applications in the end, which will give a wholesome touch to the abstract.
A - Dear reviewer, we very much appreciate your observation. We added information about your suggestion on the abstract.
10. Instead of using terms like ’accidents’, the authors need to use ’crashes’ as collisions do not always happen accidently, and crash/crashes is a better accepted term in the international literature and road safety organizations.
A - Dear reviewer, thank you for your suggestion. The text is fixed and revised at
this point.
Reviewer 3 Report
The manuscript is good and shows a comprehensive study. To suggest, add some discussions with a relatively similar previous study (other research papers) to be compared your findings.
Author Response
Comments and Suggestions for Authors
1- The manuscript is good and shows a comprehensive study. To suggest, add some discussions with a relatively similar previous study (other research papers) to be compared your findings.
A- Dear reviewer, we are very grateful for your comments to improve the quality of the presentation of our work. In particular, regarding its main suggestion, a table was inserted, and information from related works was discussed comparatively.
Round 2
Reviewer 2 Report
No further suggestions. The authors have incorporated all the review comments.